# Human-Biting Activity, Resting Behavior and Yellow Fever Virus Transmission Potential of *Aedes* Mosquitoes in Southwest Ethiopia

**Abate Waldetensai** [1,2,*], **Myrthe Pareyn** [3,4] **and Fekadu Massebo** [2,*]

1. Public Health Entomology Research Team, Ethiopian Public Health Institute, Addis Ababa P.O. Box 1242, Ethiopia
2. Department of Biology, Arba Minch University, Arba Minch P.O. Box 21, Ethiopia
3. Evolutionary Ecology Group, University of Antwerp, 2000 Antwerp, Belgium
4. Clinical Sciences Department, Institute of Tropical Medicine, 2000 Antwerp, Belgium
* Correspondence: abate@ephi.gov.et or abyw.res31@gmail.com (A.W.); massebofekadu@gmail.com (F.M.)

**Abstract:** Yellow fever (YF) is an emerging and re-emerging arboviral disease transmitted through the bites of infected *Aedes* mosquitoes, primarily in the genus *Aedes*. Several outbreaks of yellow fever have been documented in southern Ethiopia. Four outbreaks have been documented since 2012, suggesting that southern Ethiopia is prone to YF outbreaks. Understanding the transmission cycle is pivotal to managing arboviral disease outbreaks, and the aims of the present study were to investigate the mosquito species that most likely contributed to the recent YF outbreaks and to study their behaviors. Therefore, the present study aimed to evaluate which species of *Aedes* mosquitoes contribute to the YF virus transmission, the outbreaks that have occurred and their behaviors (biting and resting) in the region. Two districts were selected on the basis of recent YF outbreak history. A longitudinal entomological survey was conducted to collect adult mosquitoes by using human landing catches (HLC), mechanical mouth aspirators and pyrethrum sprays. Collections were conducted twice a month for six months, from February 2019 to July 2020. The mosquitoes were identified by species by using morphological keys and molecular techniques. A total of 1689 mosquitoes were collected, of which 93.7% (1582/1689) were members of the genus *Aedes* and 6.3% (107/1689) of the genus *Culex*. A total of 58.7% (991/1689) of the mosquitoes were captured in the Ofa District and 41.3% (698/1689) from the Boko Dawula District. The largest number of mosquitoes, 97.9% (1653/1689), were collected during the wet season. A total of 1582 members of the *Aedes simpsoni* complex were collected, where 57.7% (913/1582) were from the Ofa District and 42.3% (669/1582) were from the Boko Dawula District. Molecular identification showed that members of the *Aedes simpsoni* complex accounted for 99.5% (404/406), while *Aedes aegypti*, detected only in the Ofa District, accounted for only 0.5% (2/406). The mosquitoes were pooled and tested for YFV, dengue virus (DENV, serotype 1–4) and chikungunya virus (CHKV) by using qPCR. None of the 934 *Aedes simpsoni* tested were positive for any arboviruses. The human-biting activities of *Ae. simpsoni* complex were peaked between 8:00–9:00 and 16:00–17:00, mostly outdoors, both within the villages and the forests. The largest numbers of *Aedes simpsoni* complex resting mosquitoes were collected from the leaves of the Abyssinian banana, Ensete ventricosum, suggesting that they are the preferred resting places. Although the tested *Ae. simpsoni* complex was negative for arboviruses; the morning and afternoon activities of the species complex coincide with peak human outdoor activities in these areas and may therefore pose the highest risk of transmitting YFV to humans. The extremely low abundance of *Aedes aegypti* suggests a minor role in arbovirus transmission in southern Ethiopia. It is of great importance that expanded surveillance activities of arboviruses to include reservoir hosts and sylvatic vectors to the chances of devising and implementing effective control measures.

**Keywords:** *Aedes*; mosquitoes; yellow fever; *simpsoni* complex; behavior; southwestern Ethiopia

## 1. Introduction

The emergence and re-emergence of arboviral diseases raise global concerns and threats to human health [1]. The viruses usually circulate among wild animals, but spill over to humans sporadically, leading to epidemics [2]. Arboviral diseases are increasingly causing morbidity in Ethiopia, with a significant increase in the number of outbreaks in the past years. Multiple outbreaks of yellow fever (YF) have occurred and affected the non-immune human population [3]. The first YF outbreak occurred between 1960 and 1962, when 100,000 cases and 30,000 deaths were reported in the southwest of the country [3]. The second YF outbreak took place between 2012 and 2013 in South Omo in southern Ethiopia [4]. As a response, 550,000 people were vaccinated in the region [5]. In 2016, 22 suspected YF cases and five deaths were reported in South Omo again [6]. In October 2018, another YF outbreak was confirmed in Wolaita in southwest Ethiopia, resulting in the death of 10 people [7]. Following confirmation of YF outbreaks, a vaccination campaign targeting 31,565 high-risk populations was conducted in October 2018 in six identified kebeles (villages, the lowest administrative unit). In addition, a ring vaccination campaign was carried out among 1,335,865 inhabitants in nine districts, seven in the Wolaita Zone and two in the Gamo Gofa Zone [8].

Recently, the YF outbreak occurred in the Gurage Zone in southwestern Ethiopia [9]. Although YF is endemic in the country, vaccination is not part of the childhood immunization program. Therefore, children and others living outside vaccinated areas remain at risk for future YF outbreaks [6].

In addition to YF, outbreaks of other arboviral diseases, such as dengue (DEN) and chikungunya (CHIK), are also common [10]. A previous study showed the presence of *Ae. simpsoni* complex in South Omo in villages where the 2012–2014 YFV outbreaks occurred [5].

Although arboviral disease outbreaks are common in Ethiopia, limited attention has been given to entomological monitoring and surveillance to identify appropriate vector control tools for outbreak prevention and control [10]. The vector mediating the periodic outbreaks is unknown, and there is so far no evidence of whether the outbreaks are initiated by spillover from forest catchments into urban areas or other reasons [11]. Therefore, this study assessed the *Aedes* mosquito species composition, ecology, human-biting activity and role in YFV transmission in sylvatic and human occupational spaces in areas where YF outbreaks recently occurred in southern Ethiopia. This information is important to build on for further research, eventually aiming to design appropriate vector control tools which might contribute to the prevention and control of future YF epidemics.

## 2. Materials and Methods

### 2.1. Study Areas Description

This study was conducted in the Boko Dawula, former South Ari (South Omo Zone) and Ofa (Wolaita Zone) districts in southwestern Ethiopia (Figure 1). The two study districts were affected by YF outbreaks in 2016 (South Omo) and 2018 (Wolaita). The selection of the study districts was made in consultation with the zonal health office, village administrations and health extension workers. Both study sites are located on the edge of a forest, and in both districts, the village closest to the forest was selected for mosquito collection.

The Boko Dawula District is located about 20 km south of Jinka town, the capital of the South Omo Zone. The altitude ranges between 1000–1300 m above sea level (masl) [12]. The mean annual rainfall ranges between 601–1600 mm, and its mean annual temperature lies between 10–30 °C [13]. The Ofa District is located 29 km southwest of the zonal capital, Wolaita Sodo. It is located at an altitude between 1000–2000 masl [14]. The amount of rainfall is between 800 and 1400 mm, with a temperature ranging between 14–28 °C [15]. Due to high population pressure, agricultural expansion and deforestation are common. The edge of the forest is mainly used for agriculture. In forests and villages, there are natural and artificial breeding habitats for mosquitoes, such as Boko Dawula.

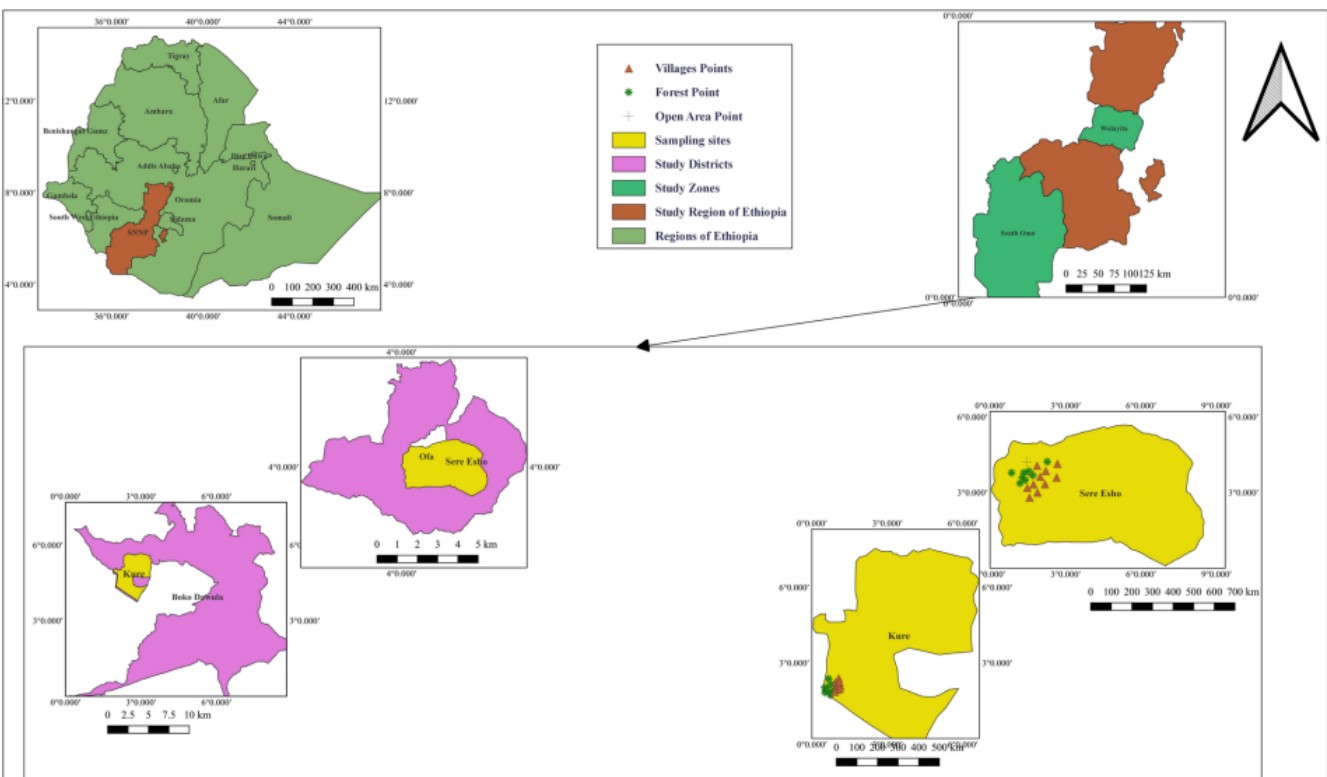

**Figure 1.** Map of the study sites in the Boko Dawula District in the South Omo Zone and the Ofa District in the Wolaita Zone, both in the Southern Nations, Nationalities and Peoples' Region (SNNPR) in southwestern Ethiopia.

## 2.2. Study Design and Sampling Techniques

We collected *Aedes* mosquitoes in three different habitats: forests (sylvatic cycle), agricultural lands located between human settlements, forests and rural human settlements (villages) for six months from February 2019 to July 2020, taking into account both the Ethiopian dry (February–April) and wet months (May–July). Human landing catches (HLC), pyrethrum space spray collection (PSC) and mechanical aspiration (MAC) were employed to collect host-seeking and resting mosquitoes.

### 2.2.1. Human Landing Collection (HLC)

Outdoor HLC was performed twice per month at three sampling habitats (villages, forests and agricultural lands) with a distance of at least 200 m between sampling sites. In the forest and agricultural land, HLC was conducted at three sites in each district. In the villages of the two selected districts, HLC was conducted both indoors and outdoors in and around five selected houses. For each house, two collectors were assigned, one indoor and one outside, with hourly rotation. The collection was performed between 6:00–10:00 and 15:00–20:00. Volunteers, wearing long-sleeve shirts, were seated in chairs with exposed legs below their knees to collect mosquitoes attempting to bite using an aspirator. Hourly, collected mosquitoes were transferred into a paper cup and preserved as described below.

### 2.2.2. Pyrethrum Spray Collection (PSC)

Indoor resting mosquito collection was conducted twice per month in fifteen randomly selected houses (other than HLC) in both villages, following the WHO indoor residual spraying (IRS) operational guideline [16]. The floors, beds and furniture were covered with white sheets, and food items were removed and properly closed before spraying an aerosol insecticide (Baygon, SC Johnson and Son Inc., Racine, WI, USA) [17]. Additionally, windows, doors and other openings were closed to prevent mosquitoes from escaping. The

collection was performed early in the morning before smoking the houses. About 10 min post-spraying, mosquitoes were collected and preserved as described below [16].

### 2.2.3. Mechanical Aspirator Collection

The MAC collection was performed twice per month at each site using a mechanical aspirator. Twelve houses were randomly selected (other than HLC and PSC) to collect mosquitoes resting on interior and exterior walls. For the collection of outdoor resting mosquitoes, four additional sites, including caves, rock holes, tree holes and leaves, were selected. A mouth aspirator, model 412 (John W. Hock Company, Gainesville, FL, USA), was used for the collection of mosquitoes. A collector spent 10 min at each site searching for resting mosquitoes using a timer. The number of mosquitoes collected and the type of resting sites were recorded. To minimize collector bias due to skill variation, collectors were alternated between collection sites.

### 2.2.4. Mosquito Species Identification

The collected mosquitoes were immediately identified at the genus level in the field using morphological keys [18,19]. *Aedes* mosquitoes were individually preserved in RNA later, transported in a cold chain and stored as soon as possible at −20 °C for further analysis. The morphologically identified *Aedes* mosquitoes were confirmed with molecular-based species identification. From all collected *Aedes* mosquitoes and identified to *Ae. simpsoni* complexes, 406 were selected for molecular species identification taking into account the habitat, collection method, time and month to make certain the representativeness of the sample. Sampling this number for molecular species identification was due to limited laboratory resources (primers and probes, reagents shortage) and budget limits. The legs of each specimen were used for DNA extraction using a NucleoSpin Tissue Kit (Macherey Nagel, Düren, Germany) according to the manufacturer's instructions. Elution was conducted in 50 µL nuclease-free water. After that, the isolated DNA was subjected to a PCR targeting the mitochondrial *cytochrome oxidase 1* (COI) gene, as described in [20].

Amplicons were checked on a 1.5% gel and sent for Sanger sequencing using the same primers. The DNA sequences were cleaned, and consensus sequences were generated and subjected to alignment using the ClustalW tool [21] implemented in MEGA × 10.1. Primers were trimmed to have sequences with equal lengths. Analyses of nucleotide composition and divergences among individuals were quantified using the Kimura 2-parameter (K2P) distance model [22]. A neighbor-joining (NJ) tree based on the K2P distances was made to provide a graphic representation of the clustering pattern among different species [23]. The presence or absence of a barcode gap was also determined for each species as a test of the reliability of its discrimination. This was to visualize and demonstrate the divergence in sequences obtained and to compare them to sequences held in GenBank. All sequences were compared with *Aedes* species sequences in the GenBank, GenBank accessions: [*Ae. simpsoni* (KT881398.1) and *Ae. aegypti* (MK300226.1)].

### 2.2.5. Arboviral Screening

For arboviral screening, mosquito specimens (head, thorax and abdomen) were pooled from 8 to 12 according to the habitat and month in which they were collected. Then, a 500 µL DNA/RNA shield (Zymo Research—Baseclear, Leiden, Netherlands) was added to each mosquito pool, and specimens were homogenized with a sterile pestle. Homogenates were centrifuged, and the supernatant was transferred to a nuclease-free tube. RNA extraction was performed using a Quick-DNA/RNA Pathogen Miniprep kit (Zymo Research–Baseclear, Leiden, Netherlands) according to the manufacturer's instructions, but the columns were replaced by Zymo-Spin IICR columns (Zymo Research). The RNA isolates were tested with three different assays: (i) a multiplex DENV 1-4 RT qPCR [20], (ii) a YFV RT-qPCR and (iii) a CHIKV RT-qPCR [24], as described before. Commercial positive and negative (nuclease-free water) PCR controls were included in each PCR to validate the run.

*2.3. Data Analysis*

Data analysis was performed in SPSS (version 21) software. Descriptive statistics (percentages, mean and standard deviations) were used to summarize the data. A generalized linear model (ANOVA), chi-squared and *t*-test were used to compare vector density and behaviors of the vectors between villages/land cover types. Human-biting rates (HBR) were measured directly from human landing collections. In all analyses, $p < 0.05$ were considered significant. The hourly mean of the biting number of adult female mosquitoes was calculated using Pearson's correlation.

**3. Results**

*3.1. Distribution of Mosquitoes*

A total of 1689 mosquitoes, 93.7% (1582/1689) of *Aedes simpsoni* complex and 6.3% (107/1689) of *Culex* were collected. Of the total collected mosquitoes, 58.7% (991/1689) are from Ofa study sites, whereas the remaining 41.3% (698/1689) are from Boko Dawula. Most of the mosquitoes were collected during the wet season, 97.9% (1653/1689). A total of 93.5% (1579/1689) of the *Aedes* complex was collected during the wet season. Of the 1582 *Aedes simpsoni* complex, 57.7% (913/1582) are from the Ofa District, and the remaining 42.3% (669/1582) are from the Boko Dawula District (Table 1).

**Table 1.** Mosquito species composition and seasonal distribution in Ofa and Boko Dawula districts, southwestern Ethiopia.

| Sites | Mosquitoes | Seasons | | | | Total | |
| | | Dry | | Wet | | | |
| | | *n* | Percent | *n* | Percent | *n* | Percent |
|---|---|---|---|---|---|---|---|
| Ofa | *Aedes* | 3 | 0.2 | 910 | 57.5 | 913 | 57.7 |
| | *Culex* | 17 | 1.0 | 61 | 3.6 | 78 | 4.6 |
| | Total | 20 | 1.2 | 971 | 57.5 | 991 | 58.7 |
| Boko Dawula | *Aedes* | 0 | 0.0 | 669 | 39.6 | 669 | 42.3 |
| | *Culex* | 16 | 0.9 | 13 | 0.8 | 29 | 1.7 |
| | Total | 16 | 0.9 | 682 | 40.4 | 698 | 41.3 |
| Total | *Aedes* | 3 | 0.02 | 1579 | 93.5 | 1582 | 93.7 |
| | *Culex* | 33 | 2 | 74 | 4.4 | 107 | 6.3 |
| Grand Total | | 36 | 2.1 | 1653 | 97.9 | 1689 | 100.0 |

A high number of *Aedes* complexes were collected from tree leaves, *E. ventricosum* Caves and interior walls during wet seasons (June to July) (Figure 2).

The majority of *Ae. simpsoni* complexes were collected by HLC (91.2%, *n* = 1442, mean = 15.02) from all land covers. Of these, 1475 (93.2%) are unfed, whereas only 107 (6.8%) are fed *Aedes* mosquitoes (Table 2).

*3.2. Hourly Biting Activity*

Of the total *Aedes* mosquitoes collected with HLC techniques (*n* = 1442) (Table 2), the hourly biting number of *Ae. simpsoni* complex significantly varies across collection times (F= 19.2; DF = 9; $p \leq 0.001$). The highest peak biting time of *Ae. simpsoni* complex is observed between 16:00 and 17:00 h (32.5/h/person). In the morning, the relative peak biting activity is reported between 8:00 and 9:00 h (19.4/h/person). The peak biting activity of *Ae. simpsoni* complex sharply declines after 10 h in the morning and 18 h in the afternoon (Figure 3).

**Table 2.** Abdominal condition of *Ae. simpsoni* complex from different collection sites by different entomological sampling techniques in southwestern Ethiopia.

| Abdominal Status | Land Cover Type | Methods | | | Total (Mean, %) |
| --- | --- | --- | --- | --- | --- |
| | | Aspiration (Mean, %) | HLC (Mean, %) | PSC (Mean, %) | |
| Fed | Forest | 41 (3.4, 2.6) | 2 (0.2, 0.1) | 0 (-, 0) | 43 (3.6, 2.7) |
| | Indoor | 1 (0.1, 0.1) | 0 (-, 0) | 0 (-, 0) | 1 (0.1, 0.1) |
| | Outdoor | 63 (5.3, 4) | 0 (-, 0) | 0 (-, 0) | 63 (5.3, 4) |
| | Total | 105 (2.2, 6.6) | 2 (0.5, 0.1) | 0 (-, 0) | 107 (2.23, 6.8) |
| Unfed | Forest | 30 (2.5, 2) | 451 (37.6, 28.5) | 0 (-, 0) | 481 (40.1, 30.4) |
| | Indoor | 0 (-, 0) | 2 (0.2, 0.1) | 3 (0.25, 0.2) | 5 (0.4, 0.3) |
| | Open Area | 0 (-, 0) | 97 (8.1, 6.1) | 0 (-, 0) | 97 (8.1, 6.1) |
| | Outdoor | 2 (0.2, 0.1) | 890 (74.2, 56.3) | 0 (-, 0) | 892 (74.3, 56.4) |
| | Total | 32 (0.7, 2) | 1440 (30, 91) | 3 (0.1, 0.) | 1475 (30.7, 93.2) |
| | Grand Total | 137 (1.43, 8.7) | 1442 (15.02, 91.2) | 3 (0.03, 0.2) | 1582 (16.5, 100) |

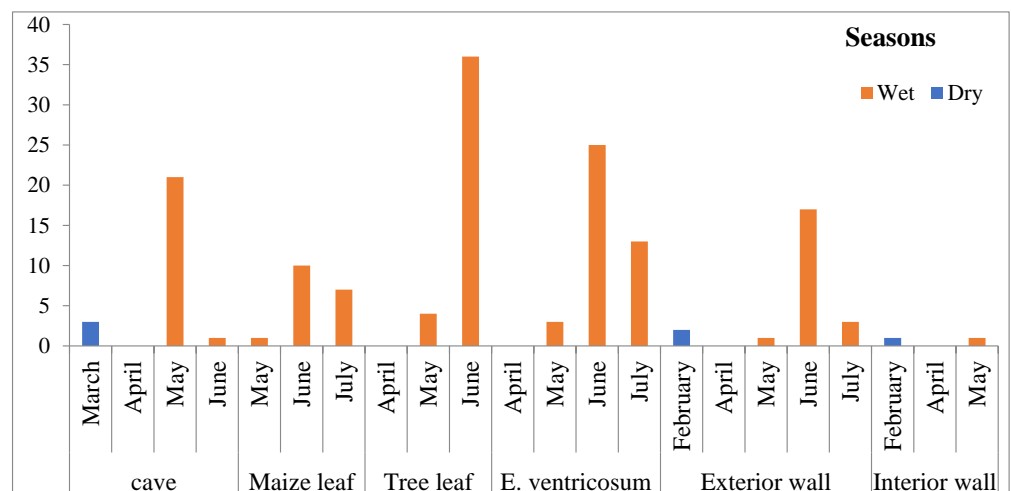

**Figure 2.** Ecological and seasonal distribution of mosquitoes in southwestern Ethiopia.

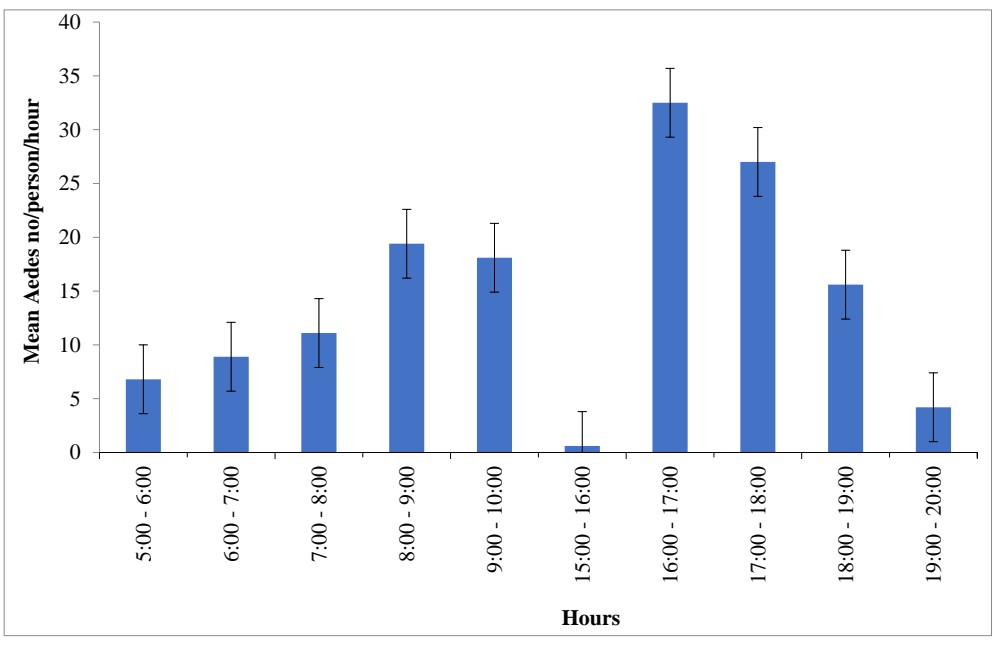

**Figure 3.** Overall hourly biting activity of *Ae. simpsoni* complex/person/h in Boko Dawula and Ofa districts, southwest Ethiopia.

The *Ae. simpsoni* complex is consistently active, with a similar pattern outdoors within the villages and inside the forests. The peak human-biting activity of *Ae. simpsoni* is observed outdoors, averaging 10.6 bites/person/h in Ofa and 6.3 bites/person/h in Boko Dawula from 16:00–17:00 h. Morning peak human-biting activity in the forest is observed from 8 to 10 am in both Ofa and Boko Dawula. The highest biting peak of *Ae. simpsoni* complex in the forest is observed between 16:00 and 17:00 h (13.4/person/h) in Boko Dawula. The peak biting times of *Aedes* is also relatively observed in open land (Figure 4).

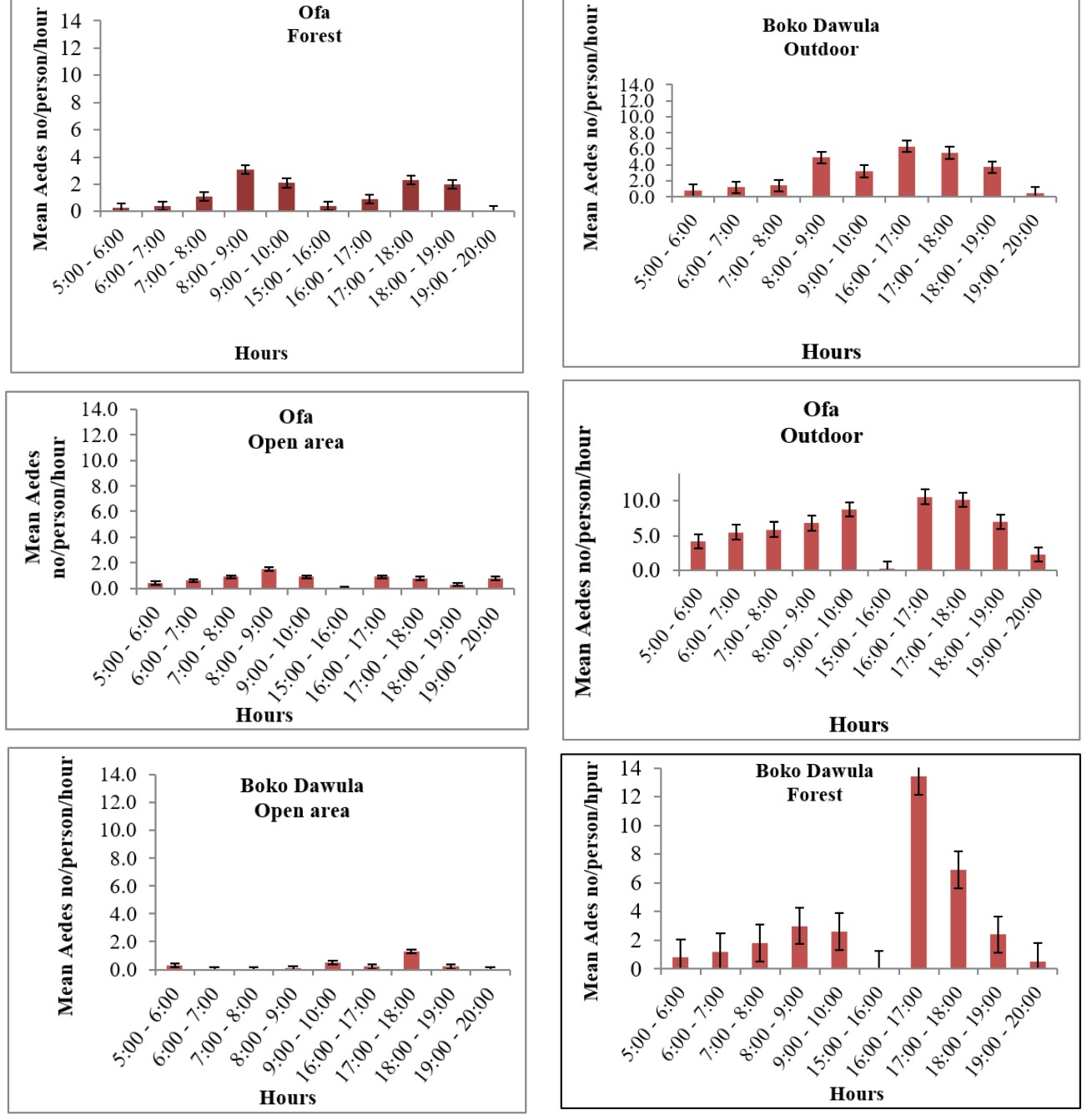

**Figure 4.** Hourly biting activity of *Aedes simpsoni* complexes/person/h in Boko Dawula and Ofa districts, southwest Ethiopia.

### 3.3. Resting Sites

To assess resting behavior, *Ae. simpsoni* complex collected by MAC was considered, by which 137 *Aedes* mosquitoes were collected overall. The density of the *Ae. simpsoni* complex varies by resting sites ($p < 0.001$, df = 11). Relatively larger numbers of *Ae. simpsoni* are collected from *E. ventricosum*, followed by tree leaves (Table 3). A single *Ae. simpsoni* complex is collected on the interior wall. Overall, most *Ae. simpsoni* are collected during the rainy months of May, June and July.

**Table 3.** Densities of *Aedes simpsoni* complex from resting sites by mouth aspirators in southwest Ethiopia.

| Collection Sites | February | March | April | May | June | July | Total | Median | IQR ** |
|---|---|---|---|---|---|---|---|---|---|
| Cave | 0 | 3 | 0 | 15 | 1 | 0 | 19 | 0.5 | 2.5 |
| *E. ventrilcosum* | 0 | 0 | 0 | 3 | 25 | 13 | 41 | 1.5 | 10.5 |
| Maize Leaf | 0 | 0 | 0 | 0 | 10 | 7 | 17 | 0 | 5.25 |
| Leaves | 0 | 0 | 0 | 3 | 35 | 0 | 38 | 0 | 2.25 |
| Interior Wall | 0 | 0 | 0 | 1 | 0 | 0 | 1 | 0 | 0 |
| Exterior Wall | 0 | 0 | 0 | 1 | 17 | 3 | 21 | 0.5 | 2.5 |
| Total | 0 | 3 | 0 | 23 | 88 | 23 | 137 | 13 | 22.25 |
| Total Median | 0 | 0 | 0 | 2 | 13.5 | 1.5 | 20 | 0.75 | 1.875 |
| IQR ** | 0 | 0 | 0 | 2 | 19.75 | 6 | 16.25 | 1 | 5 |

IQR: ** Interquartile range.

### 3.4. Molecular Speciation of Aedes Mosquitoes

Of the 406 *Aedes* (198; 48.8% from South Ari and 208; 51.2% from Ofa) mosquitoes screened for species identification, 404 (99.5%) are the *Ae. simpsoni* complex. The other 404 *Aedes* mosquitoes have more than 84% closeness to the *Ae. simpsoni* (Table 4).

**Table 4.** *Aedes* species distribution and abundances in South Ari and Ofa districts, southern Ethiopia (February–July 2019).

| Collection Sites | Study Sites | | | Overall | | Total (%) |
|---|---|---|---|---|---|---|
| | South Ari | Ofa | | | | |
| | *Ae. simpsoni* Complex (%) | *Ae. aegypti* (%) | *Ae. simpsoni* Complex (%) | *Ae. aegypti* (%) | *Ae. simpsoni* Complex (%) | |
| Outdoor | 62 (15.3) | 2 (0.5) | 92 (22.7) | 2 (0.5) | 154 (37.9) | 156 (38.4) |
| Indoor | 0 (0) | 0 (0) | 2 (0.5) | 0 (0) | 2 (0.5) | 2 (0.5) |
| Forest | 136 (33.5) | 0 (0) | 112 (27.6) | 0 (0) | 248 (61.1) | 248 (61.1) |
| Total | 198 (48.8) | 2 (0.5) | 206 (50.7) | 2 (0.5) | 404 (99.5) | 406 (100) |

Of the species identified through amplification of mitochondrial gene COI, two *Ae. aegypti* have 98% proximity to *Ae. aegypti* in the GenBank. The other 404 *Ae. simpsoni* complexes have more than 84% closeness to the *Ae. simpsoni*. All were coded and identified as being very close to the mosquitoes in the GenBank, and a phylogenetic tree was made. Of the species identified through amplification of mitochondrial gene COI, two *Ae. aegypti* have 98% proximity to *Ae. aegypti* in the GenBank. The other 404 *Ae. simpsoni* complexes have more than 84% closeness to the *Ae. simpsoni* from GenBank. All were coded and identified as being very close to the mosquitoes in the GenBank, and a phylogenetic tree was made (Figure 5). Of the 934 *Ae. simpsoni* complex screened for viral infection by RT-qPCR, none of the pools are positive for YFV, DENV, serotype 1–4 or CHKV.

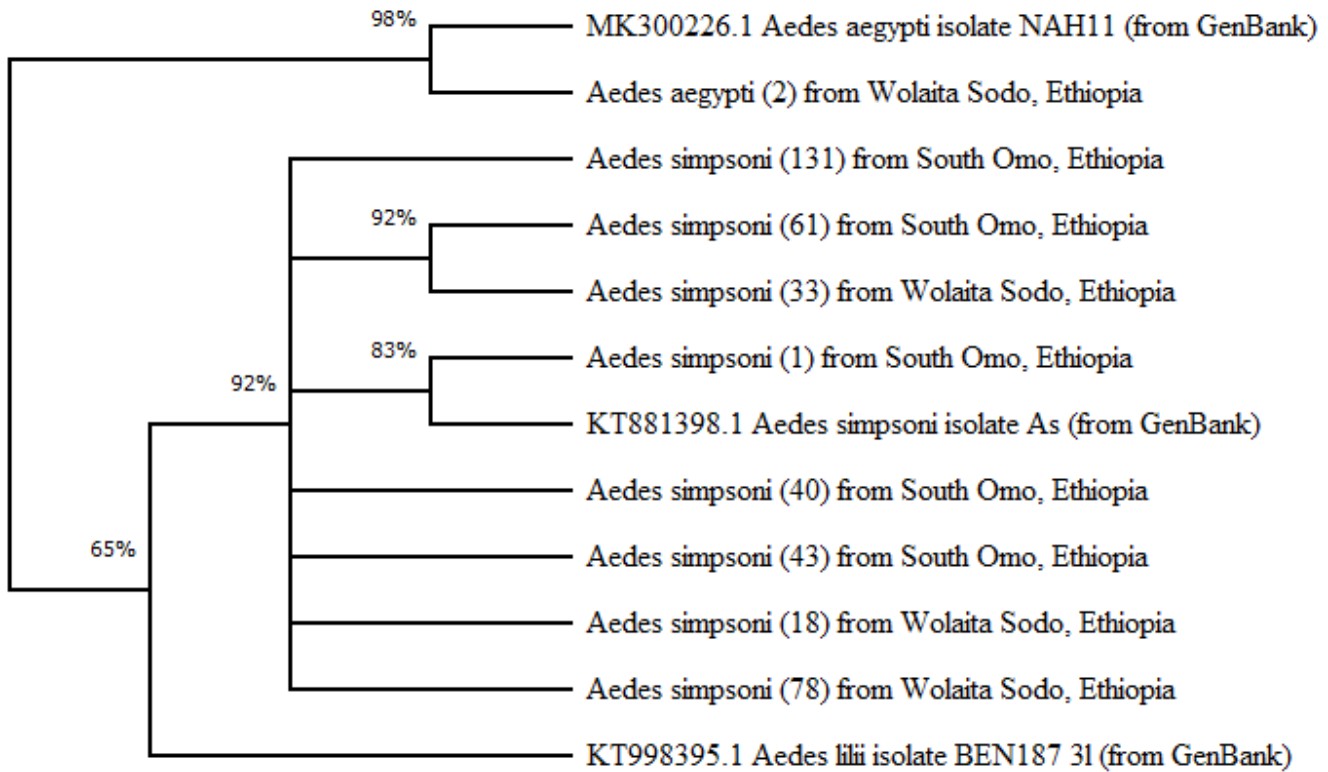

**Figure 5.** Evolutionary analysis by the evolutionary likelihood method and the Tamura-Nei model: a phylogenetic tree. The evolutionary history was inferred using the neighbor-joining method [25]. The optimal tree with the sum of branch length = 0.17579417 is shown. The percentage of replicate trees in which the associated taxa clustered together in the bootstrap test (500 replicates) is shown next to the branches. The evolutionary distances were computed using the Maximum Composite Likelihood method [20] and are in the units of the number of base substitutions per site. The differences in the composition bias among sequences were considered in evolutionary comparisons. The proportion of sites where at least one unambiguous base is present in at least one sequence for each descendant clade is shown next to each internal node in the tree. This analysis involved 12 nucleotide sequences. All ambiguous positions were removed for each sequence pair (pairwise deletion option). There were a total of 724 positions in the final dataset. Evolutionary analyses were conducted in MEGA X [23].

## 4. Discussion

The current study was conducted to assess the *Aedes simpsoni* complex distribution, human-biting activity and their role in viral transmission in selected districts with a previous history of YF outbreaks in southwestern Ethiopia. In the current study, a high number of mosquitoes were collected during the wet season. This implied that mosquito abundance might be correlated with the increased availability of natural and artificial sites for mosquito breeding and development. Differences in abundance between the wet and dry seasons may impact the transmission risk of YFV. Coincidentally, mosquito diversity and species richness showed a significant variation across the sampling periods and correlated with the period of rainfall presence [26]. Seasonal impact of temperature, relative humidity and nutrient sources contribute to mosquito abundance during the rainy season [26], while the absence of all or any of these components and high or extreme temperature decreases mosquito breeding and plenitude or abundance in the dry season [27].

The *Ae. simpsoni* complex was the dominant *Aedes* mosquito species in the two districts, followed by *Culex* mosquitoes. In the current study sites, there are no previously conducted biting behavioral studies on *Aedes* mosquitoes. The species was predominantly captured outdoors, particularly for human-biting activity. The majority of the *Aedes* mosquito was collected outdoors and within the forest. Very rarely were *Aedes* mosquitoes collected indoors. This indicated that *Aedes* mosquitoes are commonly biting humans outdoors and within the forests. The current observation supports the finding of [28], who reported larger numbers of *Aedes* biting outdoors than indoors [29]. In accordance with the study conducted in southeastern Senegal by [28], the outdoor biting behavior of *Aedes* might lead to an increase in the risk of exposure to YF transmission outdoors than indoors.

The *Aedes simpsoni* complex, specifically, *Ae. bromeliae* and *Ae. aegypti*, have both been previously documented as the responsible vectors for the re-emergence of the yellow fever outbreak in Boko Dawula, southwest Ethiopia [3]. Adult mosquitoes were collected both indoors and outdoors in South Omo (Boko Dawula) and identified as *Ae. simpsoni* complex [5]. The *Ae. simpsoni* complex was the predominant species in these previous studies, which may imply that the species has adapted to the sylvatic environment. In contrast to our study, in the previous study of [5], adult mosquitoes collected from South Omo were most likely *Ae. lilii*. In contrast, *Ae. aegypti* was dominant in Dire Dawa town, an urban setting in the eastern part of the country where YF outbreaks previously occurred [11]. This might be due to differences in the sylvatic and urban environments, and the species distribution could vary due to the adaptability or the species' ability to adapt to particular environments. The same study screened the *Ae. simpsoni* complex for arboviral infection and none of them were positive for YFV and other arboviruses. The absence of virus circulation in the *Aedes* mosquito population may be due to the absence of the movement of viremic vertebrates and/or infected mosquitoes [27]. This finding suggests that the presence of high abundance of susceptible *Aedes* mosquitoes and their feeding behavior may not be sufficient in estimating the risk of YF transmission. The differential proximity of *Aedes* to human dwelling/activity may be a contributing factor to the differential epidemiology and outbreak pattern of YF in the different environmental variables [26]. According to the study report in Kenya, in addition to vector distribution, virus presence and intrinsic and extrinsic factors contributed to viruses distribution, influencing the vectorial capacity of mosquitoes and presence of amplifying vertebrate hosts [26,27].

The majority of the host-seeking *Ae. simpsoni* complex mosquitoes were collected outdoors within the village and forest. The tendency of an *Ae. simpsoni* complex to bite humans inside the forest and outdoors within the village might increase human exposure to YFV infection when they go into the forests to cut trees for agricultural expansion and other activities. Moreover, this behavior of the *Ae. simpsoni* complex might make them suitable as a vector for the transmission of YFV in the sylvatic cycle. The current observation supports the finding of [28], who reported larger numbers of *Aedes* biting outdoors than indoors. This finding may depict that vectors highly invade villages from the surrounding land cover types, and the edge of villages may be at high risk of infection, in line with the study report of [26].

The human-biting activity of *Ae. simpsoni* was found to be high in the afternoon, with peak biting time at 16:00–17:00 h both within the villages and forests. Peak biting times are usually altered to times when their potential hosts are less protected [30]. This indicates the highest risk of coming into contact with an *Ae. Simpsoni complex* is during outdoor activities in the afternoon. However, the mosquitoes are also quite active in the morning, when lots of human activities take place in the forests, as it is generally colder then. This might increase the risk of infection due to high human vector contacts [9,30].

The *Aedes simpsoni* complex preferred to rest outdoors under the leaves of *E. vetricosum* and other plants. *E. vetricosum* was previously reported as the main breeding habitat for the same species in southern Ethiopia [3,5]. This plant has a stem resistant to evaporation and stores water for a long time, which makes it ideal for mosquito breeding and resting [31]. This implies that the *E. vetricosum* could be a potential site for monitoring and surveillance of both immature and adult stages.

None of the *Ae. simpsoni* complexes screened for arboviral diseases were found to be positive. A few specimens of the same species from South Omo (South Ari) have been previously screened for different arboviruses, and none of them were also positive [5]. The absence of virus circulation in this species might be due to the small number of mosquitoes screened or the absence of the movement of viremic vertebrates. Moreover, the YF virus transmission depends on the density of mosquitoes and wild primates [32]. During the outbreak of YF in the two districts, the majority of the community was vaccinated, which might also contribute to the absence of the virus in the vector population [30].

Additional to the *Ae. Simpsoni* complex, our study also found two *Ae. Aegypti* mosquitoes in village habitats during afternoon times with the outdoor HLC method. In contrast to the current study, a high abundance of *Ae. aegypti* was reported in Dire Dawa, eastern Ethiopia, by [10] from an immature stage collection. This indicates that the *Ae. aegypti* complex is exophagic, in parallel to the study conducted in Northern Ghana, which confirmed the *Ae. Aegypti*'s outdoor biting tendency [30]. Similarly, the study from south-eastern Senegal documented the outdoor biting tendency of *Ae. aegypti* [33]. Similar reports in Northern Ghana show that adult *Aedes spp.* are diurnal feeders with exophilic feeding behavior during the early or late hours of the day, usually biting and resting outdoors before and after feeding [30].

Despite regular outbreaks of YF in Ethiopia, very little is known about the vectors and their behavior. However, this information is pivotal to designing vector control tools to prevent and control future YF epidemics. Our research findings give new insight into the temporal and spatial dynamics of the unusual *Aedes* mosquito species as vectors of arboviral disease transmission and its amplifications in their animal reservoirs that result in spillover infection of humans in an area. While many vectors may participate in the maintenance of sylvatic arboviral diseases, the *Ae. simpsoni* complex is most likely to be responsible for the spillover into humans due to its broad land cover preferences and rates of human contact within village perimeters. This information can be used to inform the local population of the places and times of greatest risk for exposure so that mosquito avoidance or protective measures can be implemented.

There are several weaknesses in this reported study's design, which need to be considered when interpreting the results of the current study data. Moreover, the current study has not covered large geographical areas to diversify the species and obtain positive arboviruses. As this study was limited to limited study sites, it would not have been appropriate to perform a study on non-human primates and mosquito species diversity for the detection of arboviruses. This study also did not determine the parity or longevity and infectivity rates with mosquito dissections due to the physiological *Aedes*' nature and the high number of unfed mosquitoes collected in the field. Regarding entomological indices, immature stage collection for entomological risk analysis was not conducted, and it was not possible to assess the relative importance of each breeding site over time in the *Aedes* mosquito's species distribution factors (temperature, humidity and rainfall) analysis. As this study employed the use of only three collection techniques, it would not have been appropriate to collect a high number of blood-fed mosquitoes for blood meal origin analysis.

## 5. Conclusions

The *Aedes simpsoni* complex was the most dominant species in the region and could be a potential vector responsible for the transmission of YFV. The human-biting activity of *Ae. simpsoni* complex was predominantly in the morning (8:00–9:00 h) and afternoon (16:00–17:00 h) both outdoors within the villages and inside the forests where people are active. Screening more mosquito samples is recommended to identify the most important vectors mediating transmissions. Future research should focus on sylvatic vector distribution at large geographic areas and blood meal sources, including reservoirs, using diversified trap types to collect *Aedes* before, during the outbreak and after the outbreak. Assessing the blood meal sources for a comprehensive assessment of arbovirus risk and estimating the reservoir hosts of YFV should be required.

**Author Contributions:** Conceptualization, A.W. and F.M.; methodology, A.W., M.P. and F.M.; resources, A.W. and F.M.; investigation, A.W.; visualization, A.W., M.P. and F.M.; validation, A.W., M.P. and F.M.; data curation, A.W.; formal analysis, A.W., M.P. and F.M.; writing—original draft, A.W., M.P. and F.M.; writing—review and editing, A.W., M.P. and F.M.; supervision, M.P. and F.M.; project administration, F.M. All authors have read and agreed to the published version of the manuscript.

**Funding:** The Norwegian Programme for Capacity Development in Higher Education and Research for Development, Arba Minch University (ETH-13/0025), funded this work. The funders played no role in the study design, data collection and analysis or preparation of the manuscript.

**Institutional Review Board Statement:** Ethical clearance was obtained from the Arba Minch University Institutional Review Board (reference No. IRB/108/11).

**Informed Consent Statement:** Local administrative bodies and district health offices were informed and involved in the study. Written and oral consent was obtained from the study participants (households and data collectors). For entomological collections, YF-vaccinated volunteers were included and provided with anti-malarial prophylaxis.

**Data Availability Statement:** All relevant data within the manuscript are available online at figshare—My data.

**Acknowledgments:** We are very grateful to the South Omo Zone and Wolaita Zone's health department office leaders, the village head, the health extension workers and the volunteers for their cooperation during data collection. We are so proud to express our deep gratitude to the data collectors for their strong commitment to collecting mosquitoes. Our special thanks go out to Girum Tamiru (Arba Minch University) and Natalie van Houtte of the Evolutionary Ecology Group (University of Antwerp, Belgium) and Joachim Mariën from the Outbreak Research Team (Institute of Tropical Medicine Antwerp, Belgium) for their technical assistance in the laboratory process and activities.

**Conflicts of Interest:** The authors declare no conflict of interest.

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
