# Peer review of "Human-Biting Activity, Resting Behavior and Yellow Fever Virus Transmission Potential of Aedes Mosquitoes in Southwest Ethiopia"

_parasitologia, doi:10.3390/parasitologia3010011_

Round 1

Reviewer 1 Report

This is a well-planned and well executed study. I have some comments on the attached review summary document.

Author Response

Point 1: There is no list or lists of mosquito species collected. Usually during this type of studies, a variety of mosquito species is collected, and it is important to include this list to get an idea of the species composition at a particular site and to illustrate the point that other potential YFV vectors were not in the area. In this case since some of the studies were conducted in the forest and human landing catches were used, we would expect the species captured to include Aedes africanus the primary sylvatic YFV vector in sub Saharan Africa.

Response 1: During the study periods in the sampled study areas, only Aedes simpsoni complex and Culex were collected in all sample collection sites. Yes, it is great scientific idea raised by the reviewer, we couldn’t get any other aedes species (Ae. africanus) during the collection periods, may be due to limited adult collection techniques. We accept the comment and we have described limitation of this study in study limitation section.

Point 2: The use of the term Aedes is somewhat confusing. There are more that 900 species of mosquitoes in the genus Aedes worldwide. I would suggest the authors write the full names of the specific Aedes species they are discussing. That should include the Culex species as well.

Response 2: Comment accepted, and manuscript revised.

Point 3: Presentation the authors should work on the presentation to improve clarity. I have modified two sections as an example, but I cannot re-write the entire manuscript.

Response 3: Comment accepted, and manuscript revised.  

Point 4: The authors concluded that the Abyssinian banana, Ensete ventricosum, may be the preferred resting site for Ae. simpsoni complex. This complex is known to exploit phytotelmata including leaf axils of several plants such as Colocasia plants and some types of plantains. I wonder if the leaf axils of the Abyssinian bananas were not the primary larval habitats of the Ae. simpsoni complex in southern Ethiopia.

Response 4: The comment accepted, but one of the limitations of this study was collection of immature stage to determine most potential breeding sites and ecological variation which was described under the limitation section.

Point 5: Aedes mosquitoes are the primary vectors for a wide variety of arboviruses of public health and veterinary importance. In Ethiopia, yellow fever (YF) outbreaks are frequent, and they claim the lives of a substantial number of people. Therefore, understanding local transmission cycles and transmission dynamics are crucial to designing prevention and control strategies. We sampled mosquito populations in two districts in southwestern Ethiopia; these two districts had experienced recent YF outbreaks. Our aim was to identify and characterize the behavior and ecology of the primary yellow fever virus (YFV) vectors in these two districts. The most abundant mosquitoes captured in both districts were members of the Aedes (Stegomyia) simpsoni complex which includes YFV vectors. The biting activity of the Ae. simpsoni complex was bimodal with a one peak in the morning and the other peal in the afternoon; interestingly that is when most people are active outdoors within the villages and forests. None of the tested Ae. simpsoni complex mosquitoes were positive for arboviruses, suggesting that there were no YFV or extremely low and undetectable levels of YFV circulating during our sampling period. However, YFV activity surveillance can be augmented by monitoring sylvatic vectors and vertebrate hosts especially sylvatic primates

Response 5: The comment accepted; the manuscript revised.

Point 6: The Abstract Lines 21 52.

Response 6: The comment accepted; the manuscript revised.

Thanks

Reviewer 2 Report

I describe in the paragraphs below some considerations and observations regarding the article entitled: “Human biting activity, resting behavior and yellow virus transmission potential of Aedes mosquitoes in southwest Ethiopia”.

Studies such as this one are fundamental, aiming at distinguishing mosquitoes that are members of a species complex such as Ae.simpsoni s.l. Different species within a complex or group may exhibit different ecological aspects, hourly activity for biting or searching for the host, seasonality, frequency at home, parameters studied here. Evidently there are other parameters, feeding habit (anthropophilous or not anthropophilic), parity, longevity, infectivity...etc., also linked to vector capacity and competence, whose differences may point to different species with different responses to control measures.

This is a study of some ecological aspects, and not Ecology which is very broad, for some populations or complex of species of the genus Aedes, identified by molecular taxonomy.

In line 84, the authors state that among the ecological aspects they intend to study the composition of mosquitoes of the genus Aedes. However, I believe that there must be other species for this genus occurring in the two study sites, despite the authors stating in the discussion that they did not cover large geographic areas to diversify the species. I understand that the authors aimed to study for the genus Aedes the species or groups of associated species, with the possibility of vectorial activity in yellow fever.

In line 157, it is interesting to justify the non-use of morphological taxonomic keys to identify species of the genera Aedes and Culex, for later comparison and confirmation with the results of molecular taxonomy. Probably the morphological characters are so identical and superimposed that they make the correct morphological identification difficult.

Regarding mechanical aspiration, line 124, it would be important to mention the type of vacuum cleaner used to collect mosquitoes in their shelters. Nasci's aspirator is widely used in collections in this type of habitat.

On line 213 the title of table 1 is incorrect. We do not have species composition in this table. We have two genera Culex and Aedes distributed during dry and wet or rainy periods.

On line 381 I believe Ae.simpsoni s.l numerically much more expressive, or more abundant compared to Ae.aegypti. I believe this is what the authors mean. There are no other identified species to make this comparison that the species complex would have been numerically dominant compared to the other species.

I liked that the authors pointed out the limits of the work. Those are the observations I make for the moment.

Author Response

Point 1: Studies such as this one are fundamental, aiming at distinguishing mosquitoes that are members of a species complex such as Ae.simpsoni s.l. Different species within a complex or group may exhibit different ecological aspects, hourly activity for biting or searching for the host, seasonality, frequency at home, parameters studied here. Evidently there are other parameters, feeding habit (anthropophilous or not anthropophilic), parity, longevity, infectivity...etc., also linked to vector capacity and competence, whose differences may point to different species with different responses to control measures.

Response 1: We accept the comment and we have described parity, longevity and infectivity with mosquitoes dissection in study limitation.

Point 2: This is a study of some ecological aspects, and not Ecology which is very broad, for some populations or complex of species of the genus Aedes, identified by molecular taxonomy.

Response 2: Thank you for the excellent sceintific justification, we understood your comment and accepted to modify the manuscript as Distribution of Mosquitoes.

Point 3: In line 84, the authors state that among the ecological aspects they intend to study the composition of mosquitoes of the genus Aedes. However, I believe that there must be other species for this genus occurring in the two study sites, despite the authors stating in the discussion that they did not cover large geographic areas to diversify the species. I understand that the authors aimed to study for the genus Aedes the species or groups of associated species, with the possibility of vectorial activity in yellow fever.

Response 3: Thank you for the excellent scientific justification, we understood your comment and accepted to modify the manuscript as Distribution of Mosquitoes. The behavioral activities of other mosquitoes like Anopheles are varies in time and places. We collected the Aedes mosquitoes during the daytime which is very different from Anopheles collection time (night) to get other diversified mosquitoes.  

Point 4: In line 157, it is interesting to justify the non-use of morphological taxonomic keys to identify species of the genera Aedes and Culex, for later comparison and confirmation with the results of molecular taxonomy. Probably the morphological characters are so identical and superimposed that they make the correct morphological identification difficult.

Response 4: The comment accepted; the manuscript revised.

Point 5: Regarding mechanical aspiration, line 124, it would be important to mention the type of vacuum cleaner used to collect mosquitoes in their shelters. Nasci's aspirator is widely used in collections in this type of habitat.

Response 5: The comment accepted; the manuscript revised.

Response 4: The comment accepted; the manuscript revised.

Point 6: In line 213 the title of table 1 is incorrect. We do not have species composition in this table. We have two genera Culex and Aedes distributed during dry and wet or rainy periods.

Response 6: The comment accepted; the manuscript revised.

Point 7: On line 381 I believe Ae.simpsoni s.l numerically much more expressive, or more abundant compared to Ae.aegypti. I believe this is what the authors mean. There are no other identified species to make this comparison that the species complex would have been numerically dominant compared to the other species.

Response 7: Thanks
